# Features and signals in precocious citation impact: A meta-research study

**John P. A. Ioannidis** [ID]*

Departments of Medicine and of Epidemiology and Population Health, Stanford University School of Medicine; and Meta-Research Innovation Center at Stanford (METRICS), Stanford University, Stanford, California, United States of America

* jioannid@stanford.edu

## Abstract

The current analysis aimed to evaluate the profiles of scientists who reach top citation impact in a very short time once they start publishing. Precocious citation impact was defined as rising to become a top-cited scientist within $t \leq 8$ years after the first publication year. Ultra-precocious citation impact was defined similarly for $t \leq 5$ years. Top-cited authors included those in the top-2% of a previously validated composite citation indicator across 174 subfields of science or in the top-100,000 authors of that composite citation indicator across all science based on Scopus. Annual data between 2017 and 2023 show a strong increase over time, with 469 precocious and 66 ultra-precocious citation impact author profiles in 2023. In-depth assessment of validated ultra-precocious scientists in 2023, showed significantly higher frequency of less developed country affiliation; clustering in 4 high-risk subfields; high self-citations for their field; being top-cited only when self-citations were included; high citations to citing papers ratio for their field; extreme publishing behavior; extreme citation orchestration metric $c/h^2$; and high percentage of citations given to first-authored papers compared with all top-cited authors ($p < 0.005$ for all signals). The 17 ultra-precocious citation impact authors in the 2017–2020 top-cited lists who had retractions showed on average 4.1 of these 8 signal indicators at the time they entered the top-cited list. In conclusion, while some authors with precocious citation impact may be stellar scientists, others probably herald massive manipulative or fraudulent behaviors infiltrating the scientific literature.

## Introduction

Some authors reach extremely high citation impact to their work very rapidly after they start publishing. These authors may include some of the very best scientists whose influential work quickly attracts major attention. Alternatively, implausible rapidly rising early citation impact at the beginning of one's career may reflect fraud or inappropriate manipulation and gaming of publications and citations.

**Data availability statement:** All the raw data are available in pre-existing publicly available databases (https://elsevier.digitalcommonsdata.com/datasets/btchxktzyw/7 and older annual versions in same site and https://elsevier.digitalcommonsdata.com/datasets/kmyvjk3xmd/2) and in the manuscript.

**Funding:** The work of John Ioannidis is supported by an unrestricted gift from Sue and Bob O'Donnell to Stanford University. The funders had no role in study design, data collection and analysis, decision to publish, or preparation of the manuscript.

**Competing interests:** METRICS has been funded by grants from the Laura and John Arnold Foundation (Arnold Ventures). This does not alter our adherence to PLOS ONE policies on sharing data and materials.

Different types of fraud and manipulations may result in precocious starts [1]. First, paper mills produce fake papers and sell authorship slots to paying authors [2–4]. Artificial intelligence may have already accelerated the production of such fake papers. Second, citation cartels may be formed among scientists who promote their work by citing each other's papers without justified referencing [5]. Third, rogue editors and coordinated fake peer review may allow some authors to publish massively bypassing any gatekeeping [6]. Fourth, plagiarism and duplication may allow publication proliferation. Fifth, scientists may engage in extreme self-citation and may even publish rubbish documents that simply boost citations [7,8]. Given that the Hirsch h-index in particular has acquired inappropriately large influence, citations may be placed strategically so as to maximize the h-index [9–11]. The ways that productivity and impact metrics can be gamed are almost endless in ingenuity. Occasionally several fraudulent and manipulative mechanisms may co-exist. Copious fake or meaningless publications may ensue. Most likely, only a minority of them get discovered and retracted – with substantial delays [12,13]. More systematic cleaning of the literature from fraudulent and meaningless papers is desirable.

Problematic behaviors may affect scientists at any career stage. However, when they occur in very early stages, the resulting pattern may be more readily recognizable. Inflation of metrics may pass unnoticed in late career, when one's work has gained momentum and is already substantially cited. Detection, conversely, may be easier in beginning careers, where it is difficult to explain how beginners rapidly rise to the extreme top unless they are rare prodigies producing extremely major contributions. Citations take time to accrue. Overtaking more senior scientists in cumulative impact may take decades of strong publication presence. It is important to develop quantitative and qualitative processes that differentiate true excellence from fraud and inappropriate practices [1,14].

The current meta-research study uses data from comprehensive science-wide citation databases which get updated annually [15]. The aim is to evaluate precocious citation impact patterns, find whether they become more frequent over time, explore their distribution across countries and scientific subfields and probe whether they are associated with any other extreme metrics that may serve as signals of potential problems in these impressive CVs.

## Methods

### Definitions

Precocious citation impact was defined here for operational purposes as rising to become a top-cited scientist within 8 (or fewer) years after the calendar year of the first publication; and ultra-precocious citation impact was defined as rising to become a top-cited scientist within 5 (or fewer) years after the calendar year of the first publication. The use of 8- and 5-year cut-offs is arbitrary. These thresholds were pre-set in the analysis so as to capture individuals who achieve extreme acceleration of their citation impact in the very early stages of their career placing them at the far end of the distribution of all scientists in this aspect. If one assumes that extremes

are populated by a mixture of individuals with truly exceptional ability and others with manipulative (or even fraudulent) behavior, it is possible that the relative share of the latter group may be enriched when an ultra-extreme threshold is used. Employing two thresholds also allowed to examine comparatively the two resulting cohorts of authors.

Top-cited was defined by membership in the list of highly-cited scientists based on career-long impact that places the author in the top-2% of a previously validated composite citation indicator [15,16] in one of the 174 subfields of science according to the Science Metrix classification [17] or in the top-100,000 authors of that composite citation indicator across all scientific subfields according to Scopus data [18]. The composite indicator is calculated from 6 metrics that consider both raw citations and h-index, but also co-authorship, and authorship placement (single, first, last). Scientists with ≥5 full papers are considered for ranking. Ranking is performed with separate calculations that include or exclude self-citations; a scientist qualifies if they pass the top-cited thresholds with either approach.

The composite indicator has been widely used in previous work [15,16] and the respective datasets have been accessed over 4 million times to date. In principle, it tries to amalgamate information on citations, co-authorship patterns, and authorship placement in positions that usually suggest greater contribution. An author is included if they manage to be in the top-2% of the composite indicator among all authors who have the same primary scientific subfield and have published at least 5 full papers. In addition, some authors are included because they are among the very top in the composite indicator across all authors in all science, even though they may not have made it explicitly to the top-2% among those who share the same primary scientific subfield. The cut-off of at least 5 full papers has been used since the original conception of the composite indicator. It aims to exclude authors with very limited presence in the literature. Articles, reviews, and conference papers are the only types of publications that count as full papers. Editorials, letters, and commentaries are not included.

## Time trends

Ranking lists of top-cited scientists have been previously published based on these thresholds annually including citations to end of 2023, 2022, 2021, 2020, and 2019. Analyses including citations to end of 2018 and 2017 are also available, but only authors who are among the top-100,000 across all science for composite citation indicator are included. All annual updated databases are available at https://elsevier.digitalcommonsdata.com/datasets/btchxktzyw/7 (versions 7,6,5,3,2,1,1 for 7 annual databases, respectively). The number of top-cited authors qualifying with first publication occurring within 1, 2, 3, 4, 5, 6, 7, or 8 past calendar years was recorded for each annually released database. Total number of top-cited authors in each annually released database allowed adjusting for increasing author numbers over time.

## In-depth evaluation of recent cohort of authors with ultra-precocious citation impact

For authors apparently meeting criteria for ultra-precocious citation impact in the most recently released database (including citation data until the end of 2023, https://elsevier.digitalcommonsdata.com/datasets/btchxktzyw/7), a more in-depth evaluation was performed. First, it was assessed whether they may have additional Scopus author ID records where early publications before 2018 (i.e., outside the 5-year frame) were included. If so, these authors were excluded, since ultra-precocious citation impact is then an artefact of Scopus inaccuracies [18]. Publication profiles of authors with verified ultra-precocious citation impact were also examined to find if any were journalists/columnists. It is well described that some journalists/columnists may publish very large numbers of items, especially in high-impact journals where even news stories and journalistic columns may rapidly attract many citations [19].

The eligible ultra-precocious scientists were then contrasted against other precocious authors (those starting to publish in 2015–2017) and the group of all top-cited authors (regardless of year of starting to publish) using the database that includes citation data to end of 2023. The following descriptive characteristics were obtained and summarized as counts for discrete variables and median for continuous ones: first publication year, country, number of papers, total

citations, Hirsch h-index and co-authorship-adjusting Schreiber hm-index (with and without self-citations), proportion of self-citations, ratio of citations to citing papers (including self-citations), ratio of citations to the square of h-index (including self-citations), percentage of citations given to first-authored papers (including single-authored ones and including self-citations), percentage of citations given to first- (including single-) or last-authored papers (including self-citations), main scientific subfield per Science Metrix, and any retractions (excluding retractions without any author error or responsibility or retractions of preprints) based on linking of the Retraction Watch Database [20] to Scopus [21]. Recorded retractions may reflect either honest error or misconduct, but misconduct seems responsible for most of them [22,23].

**Signal indicators for scientists with ultra-precocious citation impact**

For each eligible scientist with ultra-precocious citation impact in the latest updated database (those starting to publish in 2018 or later), the following signal indicators were examined that may increase the possibility that the citation impact may not reflect just extreme excellence:

1. less developed country (since it is less plausible that with fewer means a scientist may manage to achieve such extreme impact so fast); countries not classified as developed economies by the United Nations Department of Economic and Social Affairs' World Economic Situation and Prospects 2024 report were considered "less developed" (https://www.un.org/development/desa/dpad/wp-content/uploads/sites/45/WESP_2024_Web.pdf).

2. main Science Metrix subfield with significantly higher ($p < 0.005$) representation among ultra-precocious authors than among all top-cited authors (since the concentration of ultra-precocious authors in a subfield may suggest a discipline-wide problem from journals with spurious publication and citation practices).

3. proportion of self-citations exceeding the 95th percentile among all top-cited authors in the same scientific field (since this may demonstrate over-emphasis on self-promotion); broad Science Metric field was used for percentile calibration instead of subfield, because if ultra-precocious citation impact in a subfield has a strong association with extreme self-citations, this may affect the observed 95th percentile in the subfield, but any bias would be diluted at field level.

4. inclusion among the top-cited list of scientists only when self-citations are included in calculations (for same reason as above);

5. ratio of citations over number of citing papers exceeding the 95th percentile among all top-cited authors in the same scientific field (since it may suggest that too many citing papers include very large numbers of citations to that author); field-level instead of subfield-level percentile calibration was used (same reason as for the self-citations).

6. extreme publishing behavior (defined as >60 full papers [articles, reviews, and conference papers] indexed in Scopus within a single calendar year [24] which may suggest implausible peaks of productivity) – the >60 cut-off has been used as a threshold of extreme publishing behavior in previous work [24]; and

7. citations/$h^2 < 2.45$ which has been previously proposed for detection of citation orchestration [25,26], i.e., authors with very low values of this metric have a disproportionately high h-index when contrasted to their total citations and this may mean that at least some of them may have self-cited or had other scientists or even fake papers cite their work in a way that it maximizes the h-index, an index that is widely used (and misused) in research assessments [9–11]; in previous work [25], the proposed <2.45 value corresponds to the 1st percentile among all authors with 5 or more papers.

8. percentage of citations given to first-authored (including single-authored) papers exceeding the 95th percentile among all top-cited authors (the cut-off of >80.8% based on the most recent iteration of the top-cited authors' database); this indicator was added after a comment by a peer-reviewer and it heralds that an author has relatively little influential work where he is not the first or single author.

Raw data for all signal indicators were obtained from publicly available Scopus-based databases ([https://elsevier.digitalcommonsdata.com/datasets/btchxktzyw/7](https://elsevier.digitalcommonsdata.com/datasets/btchxktzyw/7) with data freeze of August 1, 2024 for country, subfield, and all citation metrics, and [https://elsevier.digitalcommonsdata.com/datasets/kmyvjk3xmd/2](https://elsevier.digitalcommonsdata.com/datasets/kmyvjk3xmd/2) for extreme publishing behavior. For each of these signal indicators, the proportion of authors with the indicator was compared between ultra-precocious citation impact authors and all top-cited scientists. Given the exploratory nature of analyses, a conservative p-value<0.005 was considered statistically significant. The threshold of <0.005 has been previously recommended as more appropriate for claiming statistical significance [27].

Then, besides the single affiliation selected by Scopus in the August 1, 2024 data freeze, for each author with ultra-precocious citation impact and a selected single affiliation from a developed country, the full publication record was assessed to identify if the author had also any affiliation(s) from less developed countries.

For each author with ultra-precocious citation impact, the total number of signal indicators was counted. Authors with >4 signals and those with <2 signals were also inspected in more depth for their work through online searches.

### Retractions among scientists with ultra-precocious citation impact in early top-cited authors' lists

Finally, for all authors recorded with ultra-precocious citation impact in any annual updated database of citation indicators over the 4 earlier available annual iterations (2017, 2018, 2019, and 2020), Retraction Watch Database was searched for any retractions (excluding retractions with clearly no error of the authors and preprint retractions), and, if so, the year of the earliest retraction, whether this was before or after their entry among the top-cited authors' list, and whether the reasons listed for the retraction included mentions of paper mills, rogue editors, fake peer review or concerns with peer review.

### Exploratory analysis statement

The analyses presented here used established, previously standardized databases, but were exploratory without pre-specified protocol. Exploration of precocious citation impact had to be iterative in probing this newly discovered, still poorly understood phenomenon.

## Results

### Overall trends over time

The number of authors with ultra-precocious or precocious citation impact has been increasing over time every year, except for some anomaly in the 2020 data (Table 1). Between 2019 and 2023, the number of Scopus author profiles with ultra-precocious citation impact increased in the annual databases from 28 to 66 (0.18 per thousand rising to 0.30 per thousand top-cited authors). The number of authors with precocious citation impact increased in the annual databases from 213 to 469 (1.3 per thousand rising to 2.2 per thousand top-cited authors). Data for 2018 and 2017 showed a very small number of authors with ultra-precocious or other precocious citation impact, but they are not directly comparable with the other years, because only scientists reaching the top-100,000 ranks across all science were included in these two years' annual lists.

### In-depth assessment of authors with apparent ultra-precocious citation impact

Of the 66 listed authors with ultra-precocious citation impact in the latest iteration of the top-cited authors' database, 2 were artefacts (they had published earlier papers erroneously placed in a different Scopus author ID file). Another 5 were journalists/columnists (2 in the BMJ, 1 in Nature and 2 in both BMJ and Nature). The remaining 59 were validated to be scientists with ultra-precocious citation impact.

**Table 1. Number of authors with precocious and ultra-precocious citation impact.**

| Year of first publication | Number of authors when citations for ranking are counted until the end of the calendar year | | | | | | |
|---|---|---|---|---|---|---|---|
| | 2023 | 2022 | 2021 | 2020 | 2019 | 2018* | 2017* |
| 2022 | 0 | | | | | | |
| 2021 | 0 | 0 | | | | | |
| 2020 | 10 | 1 | 0 | | | | |
| 2019 | 19 | 9 | 1 | 1 | | | |
| 2018 | 37 | 12 | 4 | 1 | 0 | | |
| 2017 | 61 | 28 | 13 | 10 | 1 | 0 | |
| 2016 | 119 | 61 | 24 | 22 | 5 | 0 | 0 |
| 2015 | 223 | 122 | 57 | 33 | 5 | 0 | 0 |
| 2014 | | 183 | 93 | 62 | 17 | 1 | 0 |
| 2013 | | | 115 | 90 | 28 | 4 | 0 |
| 2012 | | | | 158 | 45 | 7 | 3 |
| 2011 | | | | | 112 | 13 | 7 |
| 2010 | | | | | | 23 | 9 |
| 2009 | | | | | | | 13 |
| Total with ultra-precocious citation impact | 66 | 50 | 42 | 67 | 28 | 5 | 3 |
| Total with precocious citation impact | 469 | 416 | 307 | 377 | 213 | 48 | 32 |
| Total top-cited authors | 217097 | 204643 | 194983 | 186177 | 159683 | 105000 | 105026 |

*Data for 2018 and 2017 are not directly comparable with the other years, because only scientists reaching the top-100,000 ranks across all science were included in these two years' lists.

## Contrast of ultra-precocious, other precocious, and all top-cited authors

Descriptive characteristics for the 59 eligible scientists with ultra-precocious citation impact (starting publishing in 2018 or later) appear in Table 2, also contrasted against other authors with precocious citation impact (those starting publishing in 2015–2017) and all top-cited authors. As shown, authors with ultra-precocious citation impact were heavily enriched in affiliations from less developed countries, as compared with the list of all top-cited authors where highly developed countries, led prominently by the United States, had the lion's share.

42 of the 59 authors with ultra-precocious citation impact had affiliations from less developed countries (single affiliation chosen by Scopus in the August 1, 2024 data freeze). Affiliations in China and India were heavily enriched among authors with ultra-precocious citation impact as compared with the full list of top-cited authors. Turkey affiliations were enriched 22-fold, and several countries with only 0–0.5% representation among all top-cited scientists had 2–3 authors in the ultra-precocious cohort, i.e., 3.4–5.1% (Iraq, Iran, Ethiopia, Russia, Vietnam, Nigeria, Pakistan). Among the other precocious citation impact authors (those who started publishing in 2015–2017), there was still large enrichment in affiliations from less developed countries. In some countries (China, India, Iran, Saudi Arabia, Egypt, Saudi Arabia, Italy, Poland, Hong Kong, Bangladesh) the proportion of their representation was even higher in the other precocious cohort than in the ultra-precocious cohort.

As shown also in Table 2, number of papers and overall citation metrics tended to be lower in the ultra-precocious cohort than in the other precocious cohort and these were lower than in the full list of top-cited authors. However, the differences would be nullified and even reversed if adjusted for the number of years publishing. Self-citations and citations per citing paper were much higher in the ultra-precocious cohort than in the full list of top-cited authors and other authors were in the middle, closer if anything to the ultra-precocious cohort. The ratio of citations to the square of the h-index was much lower in the ultra-precocious cohort and in the other precocious authors than in all top-cited authors. Finally,

**Table 2. Characteristics of ultra-precocious, other precocious, and all top-cited authors (based on top-cited authors' list from citation data until end of 2023).**

|  | Ultra-precocious | Other precocious | All top-cited |
|---|---|---|---|
|  | N = 59 | N = 403 | N = 217,097 |
| Most frequent countries* |  |  |  |
| China | 8 (14%) | 66 (16%) | 10,687 (4.9%) |
| Turkey | 7 (12%) | 16 (4.0%) | 1,172 (0.5%) |
| USA | 5 (8.5%) | 53 (13%) | 84,202 (39%) |
| India | 4 (6.8%) | 37 (9.2%) | 2,939 (1.4%) |
| Iraq | 3 (5.1%) | 3 (0.7%) | 48 (0.0%) |
| Iran | 2 (3.4%) | 17 (4.2%) | 1,020 (0.5%) |
| Ethiopia | 2 (3.4%) | 0 (0.0%) | 24 (0.0%) |
| Russia | 2 (3.4%) | 5 (1.2%) | 980 (0.5%) |
| Vietnam | 2 (3.4%) | 5 (1.2%) | 74 (0.0%) |
| Nigeria | 2 (3.4%) | 6 (1.5%) | 117 (0.1%) |
| Pakistan | 2 (3.4%) | 7 (1.7%) | 223 (0.1%) |
| Canada | 2 (3.4%) | 5 (1.2%) | 9,265 (4.3%) |
| Australia | 2 (3.4%) | 12 (3.0%) | 7,448 (3.4%) |
| Egypt | 1 (1.7%) | 10 (2.5%) | 495 (0.2%) |
| Malaysia | 1 (1.7%) | 6 (1.5%) | 368 (0.2%) |
| Great Britain | 0 (0%) | 22 (5.5%) | 19,638 (9.1%) |
| Saudi Arabia | 0 (0%) | 16 (4.0%) | 675 (0.3%) |
| Italy | 0 (0%) | 15 (3.7%) | 6,271 (2.9%) |
| Poland | 0 (0%) | 12 (3.0%) | 1,244 (0.6%) |
| Hong Kong | 0 (0%) | 8 (2.0%) | 1,273 (0.6%) |
| Bangladesh | 0 (0%) | 5 (1.2%) | 68 (0.0%) |
| Number of papers, median | 77 | 95 | 162 |
| Total citations, median | 2,591 | 3,671 | 6,533 |
| Excluding self-citations | 1,956 | 2,850 | 5,637 |
| h-index, median | 27 | 31 | 40 |
| Excluding self-citations | 22 | 26 | 37 |
| hm-index, median | 12.6 | 13.9 | 18.5 |
| Excluding self-citations | 11.3 | 11.9 | 17.0 |
| % self-citations, median | 19.6 | 18.3 | 11.7 |
| Cprat**, median | 1.59 | 1.52 | 1.33 |
| Citations/(h-index)$^2$, median | 3.23 | 3.33 | 3.91 |
| % Citations to |  |  |  |
| first-authored papers, median | 45 | 46 | 24 |
| first/last-authored papers, median | 65 | 65 | 59 |
| Frequent main subfields *** |  |  |  |
| Environmental Sciences | 17 (29%) | 29 (7.2%) | 2,399 (1.1%) |
| Energy | 10 (17%) | 53 (13%) | 6,619 (3.1%) |
| Artificial Intelligence & IP | 6 (10%) | 66 (16%) | 8,479 (3.9%) |
| Mechanical Eng & Transp | 5 (8.5%) | 29 (7.2%) | 3,064 (1.4%) |
| Gen & Int Medicine | 4 (6.8%) | 24 (6.0%) | 6,889 (3.2%) |
| Materials | 1 (1.7%) | 22 (5.5%) | 6,307 (2.9%) |
| At least one retracted paper | 6 (10%) | 42 (10%) | 7,083 (3.3%) |

IP: Image Processing, Eng: Engineering. Transp: Transports

*countries are shown if they have at least 2 authors with ultra-precocious citation impact or at least 5 other authors with precocious citation impact; country affiliation is the one selected in Scopus as of a August 1, 2024 data freeze

**ratio of citations over the number of citing papers (self-citations are included)

***subfields are shown if they have at least 4 authors with ultra-precocious citation impact or at least 20 other authors with precocious citation impact.

ultra-precocious authors and other precocious authors had a larger percentage of their citations received by papers where they were first (or single) authors. The difference was less prominent when both first (including single) and last authored papers were considered (Table 2).

Scientists with ultra-precocious citation impact heavily clustered in 4 scientific subfields, Environmental Sciences, Energy, Artificial Intelligence & Image Processing, and Mechanical Engineering & Transports (p<0.005 for each of them, when their proportion in the ultra-precocious cohort was compared with the proportion among all top-cited scientists). Cumulatively, these 4 subfields accounted for 38/59 (64%) of the scientists with ultra-precocious citation impact versus only 9.5% among the full list of all top-cited scientists. These four fields were also heavily enriched among the other precocious authors (187/403, 46%). When all authors with precocious citation impact were considered, five additional fields (Building & Construction, Food Science, General & Internal Medicine, Materials, Nanomedicine, and Networking & Telecommunications) were also significantly more common (p<0.005) than among all top-cited scientists.

10% (5/59) of ultra-precocious citation impact scientists and 10% (42/406) of other precocious authors already had at least one retracted paper, which was 3-times higher than the proportion among all top-cited authors.

### Signal indicators

As shown in Fig 1, of the 59 scientists with ultra-precocious citation impact, 42 (71%) had their single Scopus-selected affiliation in a less developed country (as of August 1, 2024 data freeze), 38 (64%) had a primary subfield that was enriched in ultra-precocious scientists, 18 (31%) had self-citations exceeding the 95th percentile for their field, 12 (20%) were top-cited only when self-citations were included, 9 (15%) had a citations to citing papers ratio exceeding the 95% percentile for their field, 4 (7%) had evidence for extreme publishing behavior, and 9 (15%) had a $c/h^2$ orchestration value <2.45, and 10 (17%) had >80.8% of their citations received by first-authored (including single-authored) papers for a total of 142 signals across the 59 scientists (mean 2.4, median 2). All 8 signal indicators were significantly enriched (p<0.005) in the cohort of authors with ultra-precocious citation impact versus all top-cited authors (Fig 2), with the highest relative enrichment (30-fold) occurring for the $c/h^2$ orchestration index.

Of the 17 authors with ultra-citation precocious citation impact where the single Scopus-selected affiliation was from a developed country, perusal of the full publication record showed that 7 in fact had additional affiliation(s) from less developed countries (Iran; Turkey/Jordan; Lebanon/China; Turkey/Lebanon/Russia; India/Iran; Iran; and Hong Kong).

Four scientists had more than 4 signals, 22 had 3–4 signals, 17 had 2 signals, 14 had 1 signal, while only 2 had 0 signals. The 4 scientists with more than 4 signals had affiliations in Syria, Iran, Iraq, and Malaysia, all 4 were working in high-risk fields, 3 of them had high self-citation rates, and 3 had extreme orchestration index. Among authors with 0 signals, one peculiar situation arose with one author with 0 signals who is apparently a hospitalist. He has first-authored two narrative reviews on epidemiology of colorectal cancer and of gastric cancer in Przeglad Gastroenterologiczny that have received 1564 and 1078 citations, respectively, within 4 years, while no other paper in the long history of that journal has accumulated more than 84 citations; he has also authored another 7 highly-cited reviews on epidemiology of other specific cancer sites in World Journal of Oncology, Wspolczesna Onkologia, Medical Sciences, and Clinical and Experimental Hepatology and that are also the most-cited or second- or third most-cited papers ever in these journals.

### Retractions among ultra-precocious authors in early top-cited lists

The top-cited scientists' lists of 2017, 2018, 2019, and 2020 included a total of 103 entries of ultra-precocious authors, of which 4 were artefacts and 3 were journalists, while 10 authors appeared in two different annual lists. Of the 86 different eligible authors with ultra-precocious citation impact, 17 (20%) had at least one retraction in the Retraction Watch Database as of October 8, 2024: 3/3 (100%) in the 2017 list, 3/4 (75%) in the 2018 list, 3/27 (11%) in the 2019 list, and 10/62 (16%) in the 2020 list. For 14/17 authors with retractions, the earliest retraction had been made in a year after they had reached a top-cited list; in another case the retraction was in 2017 and the earliest available top-cited list is from 2017.

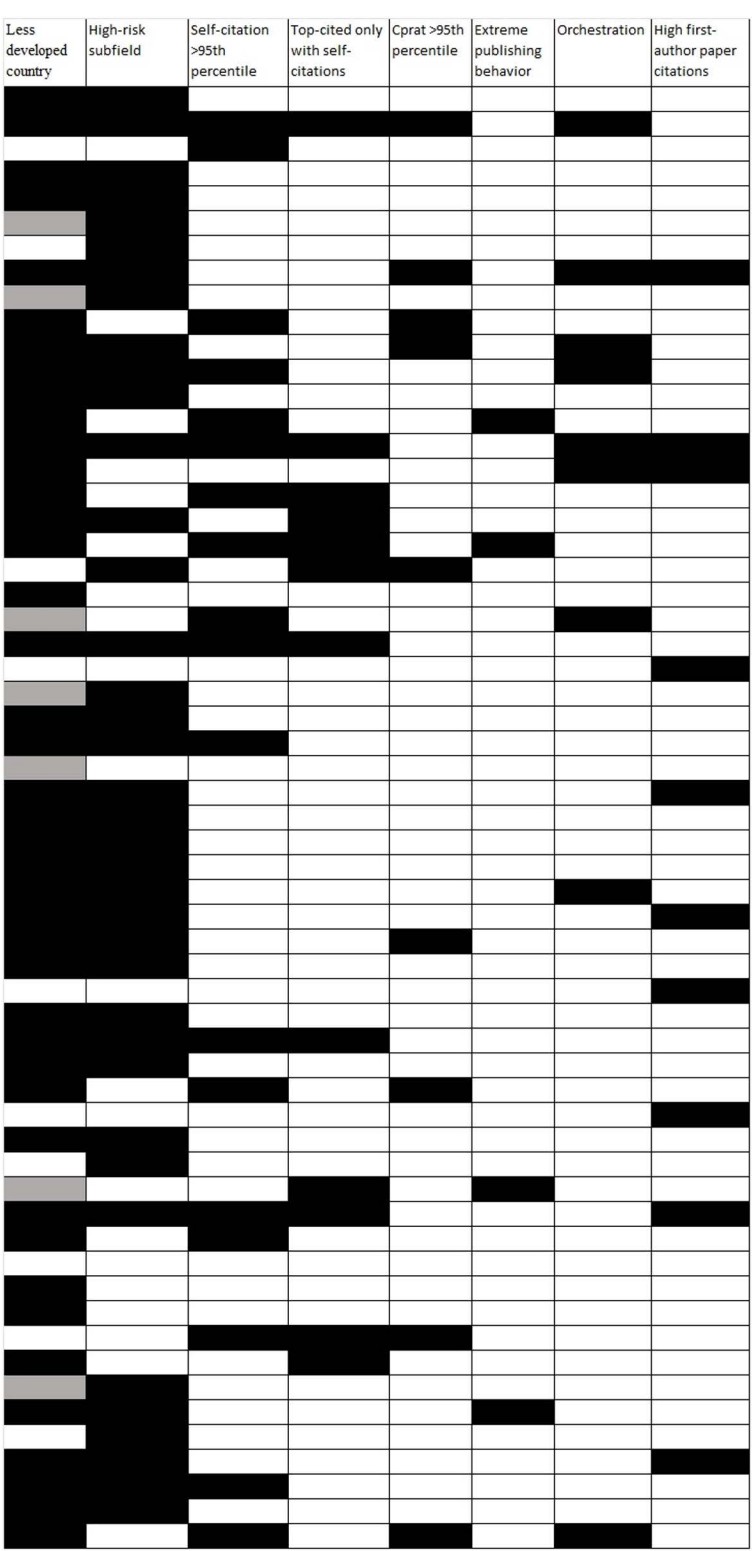

**Fig 1. Signal indicators in 59 authors with ultra-precocious citation impact among top-cited scientists based on citation counts to end of 2023.** For definitions of the 8 indicators, see Methods. Data are from Scopus and raw data can be found in https://elsevier.digitalcommonsdata.com/datasets/btchxktzyw/7 for country, subfield, and all citation metrics, and https://elsevier.digitalcommonsdata.com/datasets/kmyvjk3xmd/2 for extreme

publishing behavior. Compared with the 66 authors listed in https://elsevier.digitalcommonsdata.com/datasets/btchxktzyw/7, the figure does not include 5 journalists/columnists and 2 authors from Italy and Singapore who do not really have ultra-precocious citation impact, but represent data artefacts because Scopus split their earlier publications in a separate author file. Listing of the 59 authors here is alphabetical with one line per author. For the less developed country column, a gray color means that in the Scopus data freeze of August 1, 2024, the single author affiliation selected by Scopus for that author was from a developed country, but inspection of the full publication record of that author showed also affiliation(s) of that author with institutions in less developed countries.

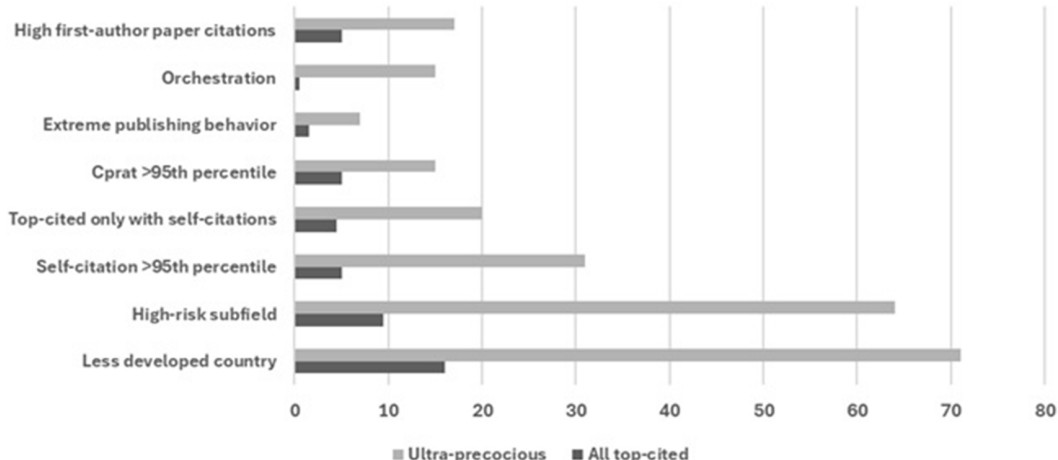

**Fig 2. Proportion of each of the 8 signal indicators in 59 authors with ultra-precocious citation impact versus in all top-cited authors.** Data are from Scopus and raw data can be found in https://elsevier.digitalcommonsdata.com/datasets/btchxktzyw/7 for country, subfield, and all citation metrics, and https://elsevier.digitalcommonsdata.com/datasets/kmyvjk3xmd/2 for extreme publishing behavior. Same selection applies as in Fig 1 for the 59 authors.

Of the 17 scientists with ultra-precocious citation impact (Fig 3), at the time of their first appearance in a top-cited authors' list 13 had their affiliation in a less developed country, 7 had a primary subfield enriched in ultra-precocious citation impact scientists, 12 had self-citations exceeding the 95th percentile for their field, 9 were top-cited only when self-citations were included, 15 had a citations to citing papers ratio exceeding the 95% percentile for their field, 3 had evidence for extreme publishing behavior, 6 had an orchestration $c/h^2$ value <2.45, and 4 had >80.8% of their citations received by first-authored (including single-authored) papers, for a total of 69 signals across the 17 scientists (mean = 4.1, median = 4). Of 4 authors with single affiliation selected by Scopus being in developed countries, 3 had additional affiliations in less developed countries (Iran; China; India).

All 17 scientists had 2 or more signals, except for one who had 0 signals (or 1 when all country affiliations were retrieved). Interestingly, that scientist currently has numerous retractions and he is widely recognized to have organized a huge number of special issues across multiple journals that cross-cited massively his work and the work of other authors thus evading detection by the self-citation indices; he also used a USA (UC Davis) affiliation, but the respective lab in that institution denies he was ever a member (https://undark.org/2023/06/21/in-a-tipsters-note-a-view-of-science-publishings-achilles-heel/).

## Discussion

Scientists with precocious and ultra-precocious citation impact have become a more frequent phenomenon over time. Such scientists emerge mostly from less developed countries, and heavily cluster in a few specific scientific subfields. Moreover, they often carry indicator signals that might reflect problematic behavior that converges

**Fig 3. Signal indicators in 17 authors with ultra-precocious citation impact among top-cited scientists based on citation counts to end of 2017, 2018, 2019, or 2020 who had retractions as of October 8, 2024 in the Retraction Watch Database (see Methods for eligible retractions).** For definitions of the 8 indicators, see Methods. Values are as of the end of the year for which they first appeared in a top-cited list. Data are from Scopus and raw data can be found in https://elsevier.digitalcommonsdata.com/datasets/btchxktzyw for country, subfield, and all citation metrics, https://elsevier.digitalcommonsdata.com/datasets/kmyvjk3xmd/2 for extreme publishing behavior. Listing of the 17 authors here is alphabetical with one line per author. For the less developed country column, a gray color means that in the Scopus data freeze of August 1, 2024, the single author affiliation selected by Scopus for that author was from a developed country, but inspection of the full publication record of that author showed also affiliation(s) of that author with institutions in less developed countries.

towards a spuriously inflated citation record. Many of these authors also have retractions, but most of the retractions occur late, after they have reached top ranks of citation metrics. Perusal of the Retraction Watch database [20] shows that paper mills, rogue editors, fake or otherwise problematic peer review are frequent reasons listed in these retractions.

Typically, when these authors first reach the levels of being top-cited in their scientific subfields they have no papers retracted yet. Apparently, they reach this extraordinary level of citation achievement extremely fast, while retractions are a slow process. It takes usually long to recognize flaws and both high-impact and low-impact journals rarely act decisively fast in retracting papers [28,29]. Therefore, perhaps several other authors in these cohorts may have papers retracted in the future. Such extraordinary CVs require more routine careful scrutiny for potential problems. The 8 signal indicators described here offer some initial indicative assessment before in-depth scrutiny of papers published by these authors and their frequent co-authors. For authors where one or a few papers are retracted, inspection of the wider published corpus and its cross-linking with other citing and cited authors and journals may reveal additional problems with papers, journals and/or author clusters. This may allow large-scale detection of problems, beyond the one-paper-at-a-time post-publication peer-review [30]. In fact, while some fraudulent authors may be working alone or with a few accomplices, currently there are fraudulent or manipulative enterprises that probably generate thousands of papers and involve also hundreds and thousands of authors, real ones or occasionally even fake ones [31,32].

The current analysis used impact ranking based on a composite citation indicator that considers 6 citation metrics addressing raw citations, co-authorship, and relative author placement [16]. This was selected in purpose instead of using just raw citations. Raw citation counts would have selected as early top-ranked scientists authoring or co-authoring single most heavily-cited papers with highest immediacy. These include mostly reference papers, guidelines, clinical trials, and some extraordinary achievements in basic and translational science. Among other simple metrics, precocious h-index

alone may also pick some problematic behaviors (e.g., strategic placement of citations [9–11,25,26]. However, it would not select preferentially the first and last authors, who may be more key players in corpora of problematic publications.

Some limitations should be acknowledged. None of the signal indicators presented here offers definitive proof of problematic behavior. However, each has substantial theoretical justification for being a risk factor and all of them are heavily enriched among the ultra-precocious cohort. Authors who reach the top within 6−8 years seem to have profiles closer to those who take 5 or fewer years than to the average top-cited scientist who takes decades. In the latest annual iteration of the top-cited scientists database, the median time elapsed since first publication is 36 years, i.e., most accomplished scientists occupy the highest citation ranks late in their careers. Therefore, one may cautiously speculate that problematic and fraudulent behavior may also exist in a substantial share of authors who show precocious but not ultra-precocious citation impact. Moreover, the definitions used here for precociousness were arbitrarily set for operational purposes to limit the authors to analyze to manageable numbers. However, problematic and fraudulent behavior may extend to other scientists regardless of whether they manage to become highly-cited and how fast they achieve this, if at all. Fraudulent and problematic behavior may far more often result in more modest inflation of CVs. This may not suffice to ever propel a CV to the top-2% of citation impact. However, the focus on precocious authors offers the advantage of studying a phenomenon at its most extreme, likely to be most heavily inclusive of problematic and fraudulent behavior at its purest forms.

It should also be stressed that many scientists with precocious citation impact may have absolutely no problematic or fraudulent behavior. They may indeed be the very exceptional early achievers who deserve only high praise for their work. Science clearly needs more early achievers and disruption unfortunately decreases over time [33,34]. The existing patterns where independence in research is reached in middle age are disappointing [35,36]. In the cohorts analyzed here, several scientists obviously fulfill a definition of meritorious excellence by diverse means of assessment besides mere citation counts, e.g., wider scientific community recognition and contributions opening new frontiers. Moreover, rapid development of highly cited emerging fields, increased prevalence of hyper-authorship practices, and large-scale collaborative projects may contribute to the emergence of ultra-precocious citation patterns. Careful scrutiny is required on a case-by-case basis, to differentiate the truly stellar scientists from those who have cut corners or even engaged in fraudulent behavior to achieve such extreme metrics so fast. In-depth assessment of single cases may also focus on assessing the citing documents, since this may reveal additional evidence for citation cartels or other questionable practices.

Each of the presented signal indicators has imperfect specificity. E.g. some scientists from less developed countries may be truly exceptional. China in particular has currently overtaken even the US in productivity and, in some fields, even in highly cited papers [37]. While some of this ascent is linked to problematic incentive structures [38, 39], elements of true excellence also exist. Moreover, resources for scientific work in less developed countries are very heterogeneous. E.g. one may compare China to Syria or Saudi Arabia to Iraq: all these countries have authors with precocious citation impact but vary vastly in resources. More importantly, non-white scientists, scientists from non-Western countries and non-native English-speaking scientists face many disadvantages [40–42] and inequities need to be corrected. Future applications of the indicators of precocious impact should be accompanied by safeguards to prevent bias or discrimination based solely on an author's geographical origin. Some truly excellent research and outstanding talent may be found in less developed countries.

Similarly, it would be unfair to dismiss all scientists working in fields that happen to have high enrichment in ultra-precocious behavior. Some of these fields are indeed cutting-edge nowadays, e.g., artificial intelligence. Furthermore, early career scientists may often legitimately have higher self-citation rates [43].

Nevertheless, the examined data suggest the presence of pervasive problems that may be currently affecting many countries with complex networks of coordinated inflation of CVs and fraud; and discipline-wide problems that may stem from journals that are less able (or less willing) to stem problematic and fraudulent papers. Indeed, retractions occur mostly in life sciences and few other scientific fields, while they are rarely used in many other disciplines [21]. Proliferation of mega-journals [44,45], predatory journals (often with difficulties in detecting or even defining them) [46–48], paper

mills, citation cartels, fake peer review and rogue editors (including also highjacked journals [49–51]) may create a perfect storm. A new landscape of massive misconduct is emerging, different from past patterns of misconduct that typically involved single, more insular authors. Illustratively, preliminary data suggest that currently one of 7 papers submitted may be fake products [52]. Estimates of fraud have been recently revised upwards [53] versus prior estimates [54], with ongoing debate about the proposed values.

The presented analyses focus on precocious citation impact as an individual author phenomenon. However, it is conceivable that the concept may be extended to research groups that extremely swiftly reach very high levels of impact. Precocious impact authors work within research teams. However, this is more difficult to study and calibrate, since there is no similar composite indicator ranking of impact for teams, as opposed to citation metrics and rankings for individuals. Nevertheless, some relevant precocious phenomena have been described for modest size teams and for entire universities. E.g. the advent of abruptly rising, massive trial publications with suspect characteristics from some center or author team [55] and the rapid growth of some institutions in number of affiliated highly cited scientists according to the Clarivate list of highly cited scientists [56].

Acknowledging the limitations of the proposed signal indicators, the analyzed data suggest that massive assessment of the scientific literature in standardized bibliometric datasets may offer evidence for widely prevalent problems. Judicious scientometric approaches [57] coupled with qualitative assessments can offer complementary insights. Learning from extreme cases may allow understanding how current and emerging problems operate and potentially how they can be more massively and more promptly detected.

## Acknowledgments

I am grateful to Kevin Boyack and Jeroen Baas for helpful discussions.

## Author contributions

**Conceptualization:** John P.A. Ioannidis.

**Data curation:** John P.A. Ioannidis.

**Formal analysis:** John P.A. Ioannidis.

**Investigation:** John P.A. Ioannidis.

**Methodology:** John P.A. Ioannidis.

**Validation:** John P.A. Ioannidis.

**Writing – original draft:** John P.A. Ioannidis.

**Writing – review & editing:** John P.A. Ioannidis.

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
