## [Decision Letter · Decision Letter 0]

Dear Dr. Ioannidis,

We look forward to receiving your revised manuscript.

Kind regards,

Miquel Vall-llosera Camps

Senior Staff Editor

PLOS ONE

Journal Requirements:

“The work of John Ioannidis is supported by an unrestricted gift from Sue and Bob O’Donnell to Stanford University.”

4. Please note that funding information should not appear in the Acknowledgments section or other areas of your manuscript. We will only publish funding information present in the Funding Statement section of the online submission form. Please remove any funding-related text from the manuscript. 

“METRICS has been funded by grants from the Laura and John Arnold Foundation (Arnold Ventures).”

6. Please note that your Data Availability Statement is currently missing the repository name and direct link to access each database. If your manuscript is accepted for publication, you will be asked to provide these details on a very short timeline. We therefore suggest that you provide this information now, though we will not hold up the peer review process if you are unable.

**Additional Editor Comments:**

I would like to sincerely apologise for the delay you have incurred with your submission. It has been exceptionally difficult to secure reviewers to evaluate your study. We have now received two completed reviews; the comments are available below. The reviewers have raised significant scientific concerns about the study that need to be addressed in a revision.

Please revise the manuscript to address all the reviewer's comments in a point-by-point response in order to ensure it is meeting the journal's publication criteria. Please note that the revised manuscript will need to undergo further review, we thus cannot at this point anticipate the outcome of the evaluation process.

Reviewers' comments:

Reviewer's Responses to Questions

**Comments to the Author**

1. Is the manuscript technically sound, and do the data support the conclusions?

Reviewer #1: Partly

Reviewer #2: Yes

2. Has the statistical analysis been performed appropriately and rigorously?

Reviewer #1: I Don't Know

Reviewer #2: Yes

3. Have the authors made all data underlying the findings in their manuscript fully available?

Reviewer #1: Yes

Reviewer #2: Yes

4. Is the manuscript presented in an intelligible fashion and written in standard English?

Reviewer #1: No

Reviewer #2: Yes

Reviewer #1: The manuscript examines precocious citation impact at the individual level, i.e., researchers who attain very high citation rates early in the career. The manuscript is overall interesting but has some issues mainly in the methods section. Many of them has to do with presentation and a lack of information and detail. Hopefully my questions and comments can help the author to improve the manuscript.

Abstract

The readability of the abstract could be increased by imposing conciseness (less detail, more substance with a focus on conclusions), better structure, and including the aim of the study.

Method

The operationalisations of “precocious citation impact” as top-cited within 8 years and “ultra-precocious citation impact” as top-cited within 5 years seems a bit arbitrary. It is not clear to me why the concept of precocious citation impact is operationalised this way? What assumptions and definitions are these operationalisations based on? My recommendation is that the author elaborate on this issue in the manuscript. (page 4) The author mentions this issue in limitations (page 15) but this should receive attention and clarification, and be discussed in the methods section.

The inclusion criteria based on “top-citedness” is not clear to me. An author is top-cited if the author is either among the top-2% of a composite citation indicator (it is unclear to me which population this percentile ranking is based on; this should be clarified in the manuscript), or among the top- 100000 authors based on the same composite citation indicator based on Scopus data. It is unclear to me how these two criteria for inclusion are related to each other. It seems as if the first is a relative cut-off threshold and the second is an absolute cut-off threshold. This makes me uncertain about to which extent these two operationalisations of “top-citedness” are comparable, and the extent to which they might refer two different definitions of top-citedness. My recommendation is that the author clarify and justify this operationalisation in the manuscript. (page 4) Overall there seem to be different operationalisations of top-citedness other than those mentioned above in the manuscript, e.g., data for 2018 and 2017 where only scientists reaching the top- 100000 ranks across all science are included (page 24). As a reader it is difficult to get a good overview of this issue, and I would recommend the author to clarify this in the manuscript. Furthermore, the use of the composite citation indicator should be justified in the manuscript. At page 14 in the discussion the indicator is justified to some degree, this should be done fully in the method section and further elaborated on.

The inclusion criteria for being considered for ranking is “Scientists with (more or equal to) 5 papers are considered for ranking”, seems arbitrary. Why is this criterion imposed and how does it impact the ranking? What does “papers” mean here? Is it only based on peer reviewed journal articles? My recommendation is that the author justify this criterion in the manuscript. (page 4)

The author write: “Total number of top-cited authors in each annually released database allowed adjusting for increasing author numbers over time” (page 5). It is not clear to me why this is done or how this is done. Can the author elaborate a bit on this?

The author write: “First, it was assessed whether they may have additional Scopus author ID records where early publications before 2018 (i.e., outside the 5-year frame) were included” (page 5). It is not clear to me how many authors where excluded due to this issue. This should be clearly stated in the manuscript. (page 5)

The author write: “The eligible ultra-precocious scientists were then contrasted against other precocious authors (those starting to publish in 2015-2017) and the group of all top-cited authors (regardless of year of starting to publish) using the database that includes citation data to end of 2023” (page 5). Who are the authors starting 2015-2017? I cannot see that these cohorts have been mentioned previously in the manuscript.

The author write: “The following descriptive characteristics were obtained and summarized as counts for discrete variables and median for continuous ones” (page 6). I cannot find this descriptitve statistics in the manuscript.

The cut-off threshold of >60 papers seems arbitrary (page 7). My recommendation is to justify this threshold with a reference of arguments.

The author write: “For each of these signal indicators, the proportion of authors with the indicator was compared between ultra-precocious citation impact authors and all top-cited scientists. Given the exploratory nature of analyses, a conservative p-value<0.005 was considered statistically significant” (page 7-8). It is not clear to me why the significance level is set at 0.005 here. My recommendation is to justify it with a reference of arguments. What is the purpose of using p-values? What is the sample (what is it’s size) and to which population do the author wish to generalize? My recommendation is to clarify these issues in the manuscript.

To summarize, overall the methods section lacks information, clarity and detail on the definitions and operationalisations of precocious citation impact and top-citedness. These issues should be presented in the manuscript and not just referenced. Further, as a reader it is difficult to follow all the references to different databases and get a clear view of what the actual dataset/datasets that is used in the study and what they consist of. My recommendation is that the author makes an effort to increase the information, clarity and detail on how the dataset/datasets that is used in this study is constructed what it consists of in the methods section in a structured and clear fashion so that the transparency on these issues increases in the manuscript.

Generel questions:

Is precocious citation impact a phenomenon that is observable first and foremost at the individual level or is it relevant also at other levels, e.g, at document level or maybe research group level, etcetera.

How does changes in the Scopus database, e.g., increased indexing over time, affect the longitudinal analyses?

This might not be possible, but, why do the author not examine the citing documents and their sources? From my point of view it would seem interesting to understand the citing side of this phenomenon.

Reviewer #2: The manuscript entitled “Features and signals in precocious citation impact: a meta-research study” addresses a highly relevant and timely topic: the emergence of patterns and signals of "precocious citation impact" and their potential associations with questionable scientific practices. The study is methodologically rigorous, employing multiple indicators to detect potential anomalies in citation trajectories. The writing is clear (although at times somewhat dense), and the overall approach is strong and systematic.

However, several aspects regarding the operationalization of key concepts, the interpretation of the findings, and the presentation of the results could be further strengthened to enhance the overall robustness, clarity, and impact of the study.

In my view, the following aspects could be further strengthened to enhance the manuscript:

- Definition of "Precocious Impact": While the selection of 8 and 5 years as thresholds for defining "precocious" and "ultra-precocious" citation impact is understandable, the manuscript would benefit from a more explicit justification of these specific cutoffs within the main text to contextualize the choice (especially, considering the diversity of fields considered in the study).

- Limitations of the signal indicators. One of the main risks associated with the use of these indicators is the potential for generalization and stigmatization. Given the notable overrepresentation of authors from less developed countries among those identified with precocious citation impact, the discussion should more explicitly address the risk of reinforcing existing structural biases or inequities in global science. Emphasizing the diversity of scientific excellence across different contexts would help provide a more balanced interpretation of the findings.

Moreover, given the sensitive nature of using country affiliation as a signal, it would be important to clearly state that any future application of such indicators in research assessments must be accompanied by strong safeguards to prevent bias or discrimination based solely on an author's geographical origin.

Now I will go briefly on more details.

Abstract

The abstract is generally clear and informative. However, I would recommend simplifying some of the longer sentences to improve readability. For example, the sentence "In-depth assessment of validated ultra-precocious scientists in 2023, showed significantly higher frequency..." is quite long and could be streamlined for clarity. Similarly, the phrase "The 17 ultra-precocious citation impact authors in the 2017-2020 top-cited lists who had retractions by October 2024..." could be simplified by removing details such as the timeline of retractions, which might be more appropriate for the main text rather than the abstract.

Finally, the transition between "top-cited authors" and "Scopus author IDs" might cause minor confusion for readers unfamiliar with the methodology; clarifying or simplifying this distinction in the abstract would enhance accessibility (population should be clear).

Methods

-Regarding the composite citation indicator, it would be helpful to clarify how co-authorship is weighted and how authorship placement (single, first, last) is specifically considered into the calculation. Additionally, when mentioning that "scientists with ≥5 full papers are considered for ranking," it would be important to specify whether all publication types (e.g., editorials, letters, commentaries) are included or if any types were excluded.

- Although the exploratory nature of the study is well justified, presenting a clearer flowchart or schematic overview of the analytical steps would greatly aid the reader in following the complex methodology.

-A brief explanation of the c/h² orchestration index within the main text would be beneficial, particularly for readers who may not be familiar with this metric.

- While it is acknowledged that none of the signal indicators provide definitive proof of misconduct, the manuscript would benefit from a more detailed discussion on the potential for false positives, particularly regarding legitimate scientists from less developed countries or rapidly emerging fields.

Results and discussion

-Figure 1 is informative, but its layout makes it somewhat difficult to interpret at a glance. Adding clearer visual separators, such as light grey horizontal lines between entries, could greatly improve readability. This would help readers more easily distinguish between different authors and the associated signal indicators.

- In the section stating "Self-citations and citations per citing paper were much higher in the ultra-precocious cohort than in the full list of top-cited authors and other authors were in the middle, closer if anything to the ultra-precocious cohort," it would be beneficial to also report authorship positions (e.g., first, last, single authorship) to better understand the roles and contributions associated with these citation patterns.

-The discussion would benefit from a more nuanced exploration of alternative explanations for ultra-precocious citation patterns. Potential factors to consider include the rapid development of highly cited emerging fields, the prevalence of international hyper-authorship practices, and the effects of large-scale collaborative projects on citation accumulation.

**Do you want your identity to be public for this peer review?** For information about this choice, including consent withdrawal, please see our Privacy Policy

Reviewer #1: No

Reviewer #2: No

---

## [Author Response · Author response to Decision Letter 1]

3 Jun 2025

May 29, 2025

Miquel Vall-llosera Camps

Senior Staff Editor

PLOS ONE

PONE-D-24-56818

Features and signals in precocious citation impact: a meta-research study

Dear Editor

I was pleased to hear that PLoS One is interested in a revised version of this manuscript. I am grateful for the insightful comments of the reviewers. I have addressed all of them in the current revision. In more detail.

Journal Requirements:

Reply: done

Reply: these have been aligned. Some funds have no grant number.

“The work of John Ioannidis is supported by an unrestricted gift from Sue and Bob O’Donnell to Stanford University.”

Reply: The funders had no role in study design, data collection and analysis, decision to publish, or preparation of the manuscript.

4. Please note that funding information should not appear in the Acknowledgments section or other areas of your manuscript. We will only publish funding information present in the Funding Statement section of the online submission form. Please remove any funding-related text from the manuscript.

Reply: Removed.

“METRICS has been funded by grants from the Laura and John Arnold Foundation (Arnold Ventures).”

Reply: This does not alter our adherence to PLOS ONE policies on sharing data and materials.

6. Please note that your Data Availability Statement is currently missing the repository name and direct link to access each database. If your manuscript is accepted for publication, you will be asked to provide these details on a very short timeline. We therefore suggest that you provide this information now, though we will not hold up the peer review process if you are unable.

Reply: As I state “All the raw data are available in pre-existing publicly available databases (see links in the text) and in the manuscript.”.

Additional Editor Comments:

I would like to sincerely apologise for the delay you have incurred with your submission. It has been exceptionally difficult to secure reviewers to evaluate your study. We have now received two completed reviews; the comments are available below. The reviewers have raised significant scientific concerns about the study that need to be addressed in a revision.

Please revise the manuscript to address all the reviewer's comments in a point-by-point response in order to ensure it is meeting the journal's publication criteria. Please note that the revised manuscript will need to undergo further review, we thus cannot at this point anticipate the outcome of the evaluation process.

Reply: Thank you for making an extra effort to secure reviewers. The comments are excellent and very helpful. Please see the point by point responses below.

Reviewers' comments:

Reviewer's Responses to Questions

Comments to the Author

1. Is the manuscript technically sound, and do the data support the conclusions?

Reviewer #1: Partly

Reviewer #2: Yes

2. Has the statistical analysis been performed appropriately and rigorously?

Reviewer #1: I Don't Know

Reviewer #2: Yes

3. Have the authors made all data underlying the findings in their manuscript fully available?

Reviewer #1: Yes

Reviewer #2: Yes

4. Is the manuscript presented in an intelligible fashion and written in standard English?

Reviewer #1: No

Reviewer #2: Yes

5. Review Comments to the Author

Reviewer #1: The manuscript examines precocious citation impact at the individual level, i.e., researchers who attain very high citation rates early in the career. The manuscript is overall interesting but has some issues mainly in the methods section. Many of them has to do with presentation and a lack of information and detail. Hopefully my questions and comments can help the author to improve the manuscript.

Reply: Thank you for the very helpful comments. Please see responses and revisions delineated below.

Abstract

The readability of the abstract could be increased by imposing conciseness (less detail, more substance with a focus on conclusions), better structure, and including the aim of the study.

Reply: the abstract has been revised accordingly.

Method

The operationalisations of “precocious citation impact” as top-cited within 8 years and “ultra-precocious citation impact” as top-cited within 5 years seems a bit arbitrary. It is not clear to me why the concept of precocious citation impact is operationalised this way? What assumptions and definitions are these operationalisations based on? My recommendation is that the author elaborate on this issue in the manuscript. (page 4) The author mentions this issue in limitations (page 15) but this should receive attention and clarification and be discussed in the methods section.

Reply: I have clarified upfront that: “The use of 8- and 5-year cut-offs is arbitrary. These thresholds were pre-set in the analysis so as to capture individuals who achieve extreme acceleration of their citation impact in the very early stages of their career placing them at the far end of the distribution of all scientists in this aspect. If one assumes that extremes are populated by a mixture of individuals with truly exceptional ability and others with manipulative (or even fraudulent behavior), it is possible that the relative share of the latter group may be enriched when an ultra-extreme threshold is used. Employing two thresholds also allowed to examine comparatively the two resulting cohorts of authors.”

The inclusion criteria based on “top-citedness” is not clear to me. An author is top-cited if the author is either among the top-2% of a composite citation indicator (it is unclear to me which population this percentile ranking is based on; this should be clarified in the manuscript), or among the top- 100000 authors based on the same composite citation indicator based on Scopus data. It is unclear to me how these two criteria for inclusion are related to each other. It seems as if the first is a relative cut-off threshold and the second is an absolute cut-off threshold. This makes me uncertain about to which extent these two operationalisations of “top-citedness” are comparable, and the extent to which they might refer two different definitions of top-citedness. My recommendation is that the author clarify and justify this operationalisation in the manuscript. (page 4) Overall there seem to be different operationalisations of top-citedness other than those mentioned above in the manuscript, e.g., data for 2018 and 2017 where only scientists reaching the top- 100000 ranks across all science are included (page 24). As a reader it is difficult to get a good overview of this issue, and I would recommend the author to clarify this in the manuscript. Furthermore, the use of the composite citation indicator should be justified in the manuscript. At page 14 in the discussion the indicator is justified to some degree, this should be done fully in the method section and further elaborated on.

Reply: I have clarified this as follows: “The composite indicator has been widely used in previous work and the respective datasets have been accessed over 4 million times to-date. In principle, it tries to amalgamate information on citations, co-authorship patterns, and authorship placement in positions that usually suggest greater contribution (single, first, senior author positions). An author is included if they manage to be in the top-2% of the composite indicator among all authors who have the same primary scientific subfield and have published at least 5 full papers. In addition, some authors are included because they are among the top-100,000 in the composite indicator across all authors in all science (roughly top-1%), even though they may not have made it to the top-2% among those who share the same primary scientific subfield.”

The inclusion criteria for being considered for ranking is “Scientists with (more or equal to) 5 papers are considered for ranking”, seems arbitrary. Why is this criterion imposed and how does it impact the ranking? What does “papers” mean here? Is it only based on peer reviewed journal articles? My recommendation is that the author justify this criterion in the manuscript. (page 4)

Reply: These criteria were set long ago on creating the composite indicator and have been extensively explained in previous work. I have nevertheless added: “the cut-off of at least 5 papers has been used since the original conception of the composite indicator. It aims to exclude authors with very limited presence in the literature. Articles, reviews, and conference papers are the only types of publications that count as papers.”

The author write: “Total number of top-cited authors in each annually released database allowed adjusting for increasing author numbers over time” (page 5). It is not clear to me why this is done or how this is done. Can the author elaborate a bit on this?

Reply: this is inherent to the selection using the top-2%, so if there are more total authors, it is still only the top-2% being selected.

The author write: “First, it was assessed whether they may have additional Scopus author ID records where early publications before 2018 (i.e., outside the 5-year frame) were included” (page 5). It is not clear to me how many authors where excluded due to this issue. This should be clearly stated in the manuscript. (page 5)

Reply: It is already stated in the Results that 2 authors were artefacts because of such split publication records.

The author write: “The eligible ultra-precocious scientists were then contrasted against other precocious authors (those starting to publish in 2015-2017) and the group of all top-cited authors (regardless of year of starting to publish) using the database that includes citation data to end of 2023” (page 5). Who are the authors starting 2015-2017? I cannot see that these cohorts have been mentioned previously in the manuscript.

Reply: Other precocious authors are those who are precocious but not ultra-precocious.

The author write: “The following descriptive characteristics were obtained and summarized as counts for discrete variables and median for continuous ones” (page 6). I cannot find this descriptive statistics in the manuscript.

Reply: this is what the Results section present, they are all descriptive statistics.

The cut-off threshold of >60 papers seems arbitrary (page 7). My recommendation is to justify this threshold with a reference of arguments.

Reply: this is a previously used cut-off, so I clarify that “the >60 cut-off has been used as a threshold of extreme publishing behavior in previous work (24”.

The author write: “For each of these signal indicators, the proportion of authors with the indicator was compared between ultra-precocious citation impact authors and all top-cited scientists. Given the exploratory nature of analyses, a conservative p-value<0.005 was considered statistically significant” (page 7-8). It is not clear to me why the significance level is set at 0.005 here. My recommendation is to justify it with a reference of arguments. What is the purpose of using p-values? What is the sample (what is it’s size) and to which population do the author wish to generalize? My recommendation is to clarify these issues in the manuscript.

Reply: A reference has been added to the classic paper (ref. 27) suggesting using 0.005 instead of 0.05. The sample here is the authors with precocious behavior versus other top-cited. These are exploratory analyses (hence 0.005 further justified), no power calculation was performed and I would avoid claiming one, this would be misleading.

To summarize, overall the methods section lacks information, clarity and detail on the definitions and operationalisations of precocious citation impact and top-citedness. These issues should be presented in the manuscript and not just referenced. Further, as a reader it is difficult to follow all the references to different databases and get a clear view of what the actual dataset/datasets that is used in the study and what they consist of. My recommendation is that the author makes an effort to increase the information, clarity and detail on how the dataset/datasets that is used in this study is constructed what it consists of in the methods section in a structured and clear fashion so that the transparency on these issues increases in the manuscript.

Reply: Thank you for the very helpful suggestions. Hopefully, with the additional referencing and clarifications, the information and clarity thereof should be improved now.

General questions:

Is precocious citation impact a phenomenon that is observable first and foremost at the individual level or is it relevant also at other levels, e.g, at document level or maybe research group level, etcetera.

Reply: I am not sure what “document level

---

## [Decision Letter · Decision Letter 1]

Features and signals in precocious citation impact: a meta-research study

PONE-D-24-56818R1

Dear Dr. Ioannidis,

We’re pleased to inform you that your manuscript has been judged scientifically suitable for publication and will be formally accepted for publication once it meets all outstanding technical requirements.

Kind regards,

Robin Haunschild

Academic Editor

PLOS ONE

Additional Editor Comments (optional):

Reviewers' comments:

Reviewer's Responses to Questions

**Comments to the Author**

Reviewer #1: All comments have been addressed

Reviewer #2: All comments have been addressed

2. Is the manuscript technically sound, and do the data support the conclusions?

Reviewer #1: (No Response)

Reviewer #2: Yes

3. Has the statistical analysis been performed appropriately and rigorously?

Reviewer #1: (No Response)

Reviewer #2: Yes

4. Have the authors made all data underlying the findings in their manuscript fully available?

Reviewer #1: (No Response)

Reviewer #2: Yes

5. Is the manuscript presented in an intelligible fashion and written in standard English?

Reviewer #1: (No Response)

Reviewer #2: Yes

Reviewer #1: (No Response)

Reviewer #2: I appreciate the substantial revisions made by the author in response to my comments. The manuscript has been significantly improved in both clarity and structure, and I commend the author for addressing the key points raised in the initial review. The methodological section has been particularly strengthened (e.g. the inclusion of a more explicit justification for the 8- and 5-year thresholds, as well as a clearer explanation of how the composite citation indicator) both of which help contextualize the rationale behind the study’s core definitions. Additionally, the discussion section now includes further clarifications that enhance the overall interpretation of the findings. In my view, the revised manuscript is now suitable for publication.

**Do you want your identity to be public for this peer review?** For information about this choice, including consent withdrawal, please see our Privacy Policy

Reviewer #1: No

Reviewer #2: No

---

## [Editor Report · Acceptance letter]

PONE-D-24-56818R1

PLOS ONE

Dear Dr. Ioannidis,

I'm pleased to inform you that your manuscript has been deemed suitable for publication in PLOS ONE. Congratulations! Your manuscript is now being handed over to our production team.

Kind regards,

on behalf of

Dr. Robin Haunschild

Academic Editor

PLOS ONE